# The Role of Latency-Associated Transcripts in the Latent Infection of Pseudorabies Virus

**DOI:** 10.3390/v14071379

**Published:** 2022-06-24

**Authors:** Jiahuan Deng, Zhuoyun Wu, Jiaqi Liu, Qiuyun Ji, Chunmei Ju

**Affiliations:** Key Laboratory of Zoonosis Prevention and Control of Guangdong Province, College of Veterinary Medicine, South China Agricultural University, Guangzhou 510642, China; 20213073021@stu.scau.edu.cn (J.D.); wzzy0615@stu.scau.edu.cn (Z.W.); 15013279249@stu.scau.edu.cn (J.L.); 20193073034@stu.scau.edu.cn (Q.J.)

**Keywords:** pseudorabies virus, latent infection, latency-associated transcripts, non-coding RNA

## Abstract

Pseudorabies virus (PRV) can cause neurological, respiratory, and reproductive diseases in pigs and establish lifelong latent infection in the peripheral nervous system (PNS). Latent infection is a typical feature of PRV, which brings great difficulties to the prevention, control, and eradication of pseudorabies. The integral mechanism of latent infection is still unclear. Latency-associated transcripts (LAT) gene is the only transcriptional region during latent infection of PRV which plays the key role in regulating viral latent infection and inhibiting apoptosis. Here, we review the characteristics of PRV latent infection and the transcriptional characteristics of the LAT gene. We also analyzed the function of non-coding RNA (ncRNA) produced by the LAT gene and its importance in latent infection. Furthermore, we provided possible strategies to solve the problem of latent infection of virulent PRV strains in the host. In short, the detailed mechanism of PRV latent infection needs to be further studied and elucidated.

## 1. Introduction

Pseudorabies virus (PRV) belongs to the family *Herpesviridae*, subfamily *Alphaherpesvirinae*, and the *Varicellovirus* genus [1]. The PRV genome is 142 kb of linear double-stranded DNA with 70 different coding genes and one latency-associated transcript (LAT) site. It consists of a unique long region (U_L_), a unique short region (U_S_), internal repetitive sequences (IRS), and terminal repetitive sequences (TRS) [2]. The natural host of PRV is pigs, but it can infect most mammals, including cattle, sheep, cats, dogs, mink, and rodents [3,4,5,6,7,8,9]. There are significant differences in PRV infection between natural and non-natural hosts [10,11]. In natural hosts, PRV can cause neurological, respiratory, and reproductive diseases and establish latent infection in the peripheral nervous system (PNS) of surviving pigs, but death in adult pigs is uncommon [11,12,13,14]. In non-natural hosts, PRV infection is characterized by severe pruritus, a short duration of disease, and rapid death [10,11]. The mortality rate of non-natural hosts is up to 100%, and consequently latent infection rarely occurs [10,11,15]. However, under laboratory conditions, PRV can establish activatable latent infection in non-natural hosts, which is of great significance in the study of herpesvirus latent infection [16,17,18,19].

PRV has been eradicated or controlled through the use of gene-deficient vaccines and differentiating infected from vaccinated animals (DIVA) strategy in many countries. However, since 2011, the emergence of mutant strains of PRV has made pseudorabies come back in China, one of the world’s largest pig breeding countries [20,21,22,23,24]. Tong et al. found that PRV mutant strain JS-2012 caused earlier clinical symptoms and higher mortality compared to PRV classic strain SC in 15, 30, and 60-day-old pigs [25]. In protection assays, the Bartha-K61 vaccine provided 100% protection against classic strain, but only partial protection against JS-2012 strain or HeN1 strain [25,26,27].The mutant strains have been prevalent in pig farms immunized with PRV vaccine, which shows that the existing PRV vaccine cannot prevent infection caused by the new PRV mutant strains [20,22,25,26,27]. In recent years, human infections of PRV have been increasingly reported. From 2018 to 2022, more than 20 cases of human PRV infection have been found [28,29,30,31,32,33,34,35,36,37,38,39]. Liu et al. isolated and identified a human PRV strain hSD-1/2019 which had high pathogenicity to mice and pigs [28]. Most of the patients infected with PRV are workers related to the pig industry, and they are directly or indirectly infected with PRV through conjunctiva, skin wounds, and syringe stab wounds. PRV is not only costly to the pig industry but also a serious threat to humans. Therefore, the eradication of PRV should be accelerated all over the world.

Latent infection is the major impediment to eradication of PRV. PRV can establish latent infection in the PNS of pigs. During the latent infection of PRV, no clinical symptoms and infectious virions exist in pigs. When stimulated by stressors, the latent virus can be reactivated, and then productive infection occurs. According to the investigation of pig farms, PRV was in a state of latent infection most of the time and latent virions were prone to reactivation in winter and spring [40]. Interestingly, although reactivated virions were detected in pigs, no clinical symptoms were observed. These virions were excreted into the environment, resulting in the spread of PRV and the infection of other susceptible animals. Therefore, controlling latent infection plays an important role in the eradication of PRV.

The integral mechanism of latent infection is still unclear. The LAT gene is the only active gene during the latent infection. It can transcribe a variety of non-coding RNAs (ncRNAs) which are involved in the establishment, maintenance, and reactivation of viral latent infection, as well as the inhibition of productive infection and anti-apoptosis [41]. Here, we mainly summarized the characteristics of PRV latent infection, the transcription and function of the LAT gene, so as to provide new perspectives for future research on PRV latent infection.

## 2. The Characteristics of PRV Latent Infection

During the natural infection of pigs, PRV replicates in the epithelial cells of the nasal mucosa and invades sensory nerve endings by membrane fusion. The virus particles entering the axon terminals are retrogradely transported to the neuron nucleus [42,43]. Then, the capsid docks near the nuclear foramen, PRV genome is released, and the tegument protein VP16 activates the immediate early gene IE180 of PRV to form a productive infection [1]. During latent infection, the expression of immediate early genes (IE gene) is affected by many factors, such as VP16, Oct-1 (a member of the Oct protein family), HCF (a cellular protein), and LAT [44]. In sensory neurons, Oct proteins (except Oct-1) can prevent the formation of VP16/Oct-1/HCF complexes, thus inhibiting the transcription of IE genes [44]. Moreover, in the nucleus, the viral genome binds to the nucleosome and further inhibits the expression of IE180 during latent infection [41]. IE180 is a potent transcriptional activator which is required for efficient transcription of early (E) gene and late (L) gene of PRV, so it is essential for viral replication [1]. The expression product of IE180 can bind to the promoter of LAT and inhibit the transcription of LAT genes [45]. Therefore, when the IE180 gene is silenced, IE180-mediated transcription of E gene and L gene is restricted, while LAT gene transcriptional activity is enhanced. Thus, LAT gene is the only transcriptional region during latent infection [46].

The PRV genome mainly exists in the nerve tissue, especially in the trigeminal ganglion, which is the most reliable tissue for detecting latent PRV during latent infection [47]. The olfactory bulb and medulla oblongata can also contain the latent genome. The PRV latent genome is mainly confined to the nucleus in the form of linear and unintegrated, and a small number of latent genomes exist in the form of ring [47]. The positive cells of latent infection are distributed in different regions of the neural tissue in the form of aggregation [47]. In latently infected neurons, the number of PRV genome is stable and is not related to the length of the latency period [47]. Latent infections are generally stable but can be reactivated under stress, such as restraint, exposure to cold, or transport [47,48].

Latent infection requires the co-regulation of viruses, neurons, and the host immune system [49,50]. When the three are balanced, the virus can establish a latent infection in the host. The conditions for the establishment of latent infection are as follows: firstly, the viral genome enters the nucleus of neurons, and the vast majority of genes are restricted for transcription and translation. Secondly, in order to avoid latent genome loss, the virus takes certain measures to promote the survival of infected cells and evade host immunity [51]. For example, LAT has an anti-apoptotic effect and can prolong the survival time of neurons [41]. Finally, latent viruses can monitor and manipulate the environment of host cells for reactivation.

## 3. Transcriptional Characteristics of LAT Gene

### 3.1. Transcriptional Region and Sizes of the LAT Gene

The study of latent viral gene expression is restricted because less than 1% of ganglion neurons contain latent viral genomes [47]. Among the latently infected neurons, Rock et al. detected DNA and mRNA of the virus by in situ hybridization, which proved that the PRV genome has transcriptional activity during the latent infection [46]. Researchers soon discovered that mRNAs produced during latent infection were transcribed from a region between 0.69 and 0.77 map units of the PRV genome [52]. This region is about 11 kb, which covers the early protein 0 (EP0) gene and IE180 gene, and the transcriptional direction is opposite to that of the IE180 and EP0 genes [53,54,55]. These mRNAs are collectively referred to as LATs, including various sizes of mRNA, in which 0.95 kb, 1.0 kb, 2.0 kb, 8.0 kb, and 8.4 kb mRNA are generally detected [52,56,57]. The mRNA of 8.4 kb (some refer to 8.5 kb) is called large latency transcript (LLT) [52,55,57,58]. Through in situ hybridization analysis of LATs, it was found that LATs were mainly confined to the nucleus of neurons and a small part of them existed in the cytoplasm [44,46].

### 3.2. The Structure and Function of LAT Promoter

The structure of the LAT promoter (LAP) overlaps with the promoter of UL1-3.5 gene cluster in the opposite direction [57]. LAP contains 2 TATA boxes, 3 CAAT boxes, and 2 GC boxes, which is a dual regulatory promoter. The first latent activation promoter (LAP1) contains the first TATA box and three CAAT boxes, and the second latent activation promoter (LAP2) contains the second TATA box and two GC boxes. LAP1 is the basic promoter of LAT gene expression during PRV latent infection. It initiates transcription of LAT in nerve tissue and produces LLT with 4.6 kb intron [58]. LLT starts at 34 nucleotides downstream of the first TATA box in LAP1 and can be spliced into different sizes of RNAs. The whole nucleotide sequence of 2.0 kb mRNA is contained in the LLT, and it lacks the intron of 4.6 kb. However, the LAT of 2.0 kb is regulated by LAP2, which starts at about 243 bp downstream of the LLT transcription initiation site and ends at the junction of *Bam*HI fragments 8′ and 8 [52]. Whether in latent or lytic infection, nerve or non-nerve cells, in vivo or in vitro, LAP2 has no specific activity, and it is responsible for regulating the transcription of 1.0 kb, 2.0 kb, and 8.0 kb LATs [52,58]. When LAP2 and LAP1 coexist, LAP2 can enhance the activity of LAP1 [57].

LAP1 mediates the transcription of LLT [58]. The nerve cells were infected by the recombinant PRV strain with a deletion of the LAP1 region in vitro. The mRNA of 2.0 kb and 8.0 kb could be obtained from the infected nerve cells, but the LLT of 8.4 kb could not be detected. The recombinant strain could establish latent infection in the pigs, but the LLT of 8.4 kb could not be detected in trigeminal nerve. Therefore, LAP1 is the key promoter of LLT, and LLT is not required for the establishment of PRV latent infection [58]. The role of LLT needs to be further understood in PRV latent infection.

LAP is neuron-specific in vivo [59,60]. Taharaguchi et al. established a transgenic mouse line containing LAP linked with the chloramphenicol acetyltransferase (CAT) gene [60]. The expression level of the CAT gene in different tissues of transgenic mice was evaluated by enzyme-linked immunosorbent assay (ELISA). It was found that CAT was almost exclusively expressed in nerve tissue, and the expression level was the highest in the trigeminal nerve [60]. The expression of CAT in trigeminal ganglion neurons was further verified by in situ hybridization. In the absence of viral proteins, LAP is not only active but also neuron-specific, indicating that LAP may be regulated by neuronal transcription factors, and is independent of viral proteins. However, in the studies by Ou and Taharaguchi et al., it has been shown that the inhibition of LAP by IE180 is caused by the formation of a stable complex of IE180, cellular protein(s), and the IE180 binding site located on LAP, suggesting that LAP can be regulated by viral and host protein(s) [45,60,61]. In order to further understand the molecular regulatory mechanism of LAP, the host protein that regulates the neuron-specificity of LAP needs to be discovered. In the study of Taharaguchi et al., there were significant differences in the expression level of the CAT gene in different neural tissues, suggesting that LAP activity varies in different neuronal environments, which may be related to the differences of neuronal transcription factors and neuronal morphology in various nerve tissues [60,62]. It has been proved that dexamethasone can activate latent infection of PRV [48,63]. When transgenic mice were treated with dexamethasone, it was found that dexamethasone did not affect LAP-mediated CAT transcription and translation. We speculate that dexamethasone induced viral reactivation may be irrelevant to LAP [60]. Therefore, it is important to analyze the interaction between LAP, viral protein, and host protein for the study of PRV latent infection.

## 4. The Role of LAT Gene in Latent Infection

Non-coding RNA (ncRNA) molecules are small and have various regulatory functions in virus replication, virus persistence, immune escape, and cellular transformation [64]. Compared with proteins, the regulation of latent infection by ncRNAs is more desirable: first, the LAT gene is the only gene with transcriptional activity during latent infection, with can transcribe a variety of ncRNAs, but does not produce bioactive proteins [65,66,67,68,69,70]; second, ncRNAs lack antigenicity and are more likely to evade host cellular immunity [64]; third, the rich functions of ncRNAs are suitable for the regulation of latent infection [64]; finally, compared with proteins, the regulatory function of ncRNAs is mild, and the regulatory mode is ideal in the setting of latent infection [71]. Therefore, the ncRNA is of great significance for the research of PRV latent infection. The LAT gene can transcribe several different types of ncRNAs, such as microRNA (miRNA), small RNA (sRNA), long non-coding RNA (lncRNA), and short non-coding RNA (sncRNA), etc. [65,66,67,68,69,70]. At present, studies on PRV latent infection are mainly focused on miRNA, and few reports on other ncRNAs. PRV LAT can transcribe many kinds of ncRNAs just like that of Herpes simplex virus type 1 (HSV-1). PRV EP0 and IE180 are homologues of HSV-1 ICP0 (infected cell polypeptide 0) and ICP4 (infected cell polypeptide 4), respectively [72]. The ncRNAs produced by HSV-1 LAT, such as miR-H2 and miR-H6, can inhibit the expression of ICP0 and ICP4 [73,74]. The ncRNAs transcribed by PRV LAT can target EP0 and IE180 mRNAs. Based on the structural similarity of homologous proteins, we speculate that some ncRNAs transcribed by PRV may have functions analogous to those of HSV. HSV’s research on LAT ncRNA is more abundant than PRV’s. Therefore, we will analyze the function of PRV LAT ncRNAs combined with HSV to comprehensively elucidate their role in latent infection.

### 4.1. MicroRNAs Transcribed by the LAT Gene and Host Cell

MicroRNA (miRNA) is about 20–24 nucleotides (nt) in size and has a variety of important regulatory functions [75]. It can regulate target mRNA by altering its stability or inhibiting its transcription. With the development of sequencing technology, the miRNAs of most herpesviruses have been identified, including PRV. However, the specific function of PRV miRNAs in the process of PRV infection is still unclear.

*Alphaherpesvirinae* has been shown to transcribe a variety of miRNAs which are usually clustered in the viral genome. The viral miRNAs are limited to LAT sites or adjacent regions and can be transcribed by each strand of the genome [72]. Anselmo et al. identified five viral miRNAs (prv-miR-LLT 1 to prv-miR-LLT 5) in PRV-infected porcine dendritic cells (DCs) by deep sequencing [76]. These miRNAs are all transcribed by the intron of LLT [76]. The sizes of prv-miR-LLT 1, prv-miR-LLT 2, prv-miR-LLT 3, and prv-miR-LLT 5 are between 21 and 23nt. Prv-miR-LLT 4 is a mature miRNA with a size of 18nt. Using gene target analysis of miRNAs (prv-miR-1,2,3,4,5), it was found that the possible targets of Prv-miR-LLT 1-5 located in LLT, EP0, and IE180. Based on Anselmo’s study, Wu et al. identified 11 viral miRNAs (prv-miR-LLT 1 to prv-miR-LLT 11) in porcine epithelial cell line (PK-15) infected with PRV by the same method [77] (Figure 1). Gene target analysis showed that prv-miR-LLT 1 and prv-miR-LLT 9 could target IE180 and LLT, and prv-miR-LLT 2 could target EP0 and LLT. It was also found that 11 viral miRNAs could target 235 host genes. GO enrichment analysis showed that these 235 host genes are involved in apoptosis, host immune response, cell metabolism, and virus replication [77]. These results suggest that viral miRNAs can play an important role in regulating the interaction between virus and host.

Although these viral miRNAs were detected during productive infection of PRV, they were produced by LAT and could regulate the latent infection of the virus. As a member of the *Alphaherpesvirinae* subfamily, the viral miRNA of PRV may have similar functions to that of HSV. MiRNAs of HSV can regulate viral latent infection by down-regulating the expression of IE genes or E genes [73,78]. In HSV-1 infected cells, LAT, as the primary miRNA precursor, can transcribe various miRNAs. Among them, HSV-1 miR-H2 can inhibit the expression of ICP0 through targeting its mRNA [73]. ICP0 is an effective activator of virus reactivation, so miR-H2 inhibits the expression of ICP0, thereby hindering virus reactivation [73,79]. When interfering with the transcription of HSV-1 miR-H2, reverse consequences will occur, including the increase of ICP0 expression, viral reactivation, and the neurovirulence of HSV [74]. Therefore, miR-H2 can regulate the conversion between latency and reactivation of the virus. In addition, HSV-1 miR-H6 can inhibit the expression of ICP4. The LAT gene of herpes simplex virus type 2 (HSV-2) can also produce many different miRNAs, among which miR-I and miR-II can reduce the expression of the neurovirulence factor ICP34.5, and miR-III can block the expression of the ICP0 gene [78,80]. MiRNAs not only regulate the latent infection, but also the productive infection. Timoneda et al. detected 8 miRNAs transcribed by LLT intron in PRV infected pigs. The expression of these viral miRNAs was obviously changed at different times of acute infection and significantly increased only in the early stage of virus infection. Thus, the authors speculated that the 8 miRNAs could be involved in the establishment of productive infection of the virus [65]. Another interesting phenomenon was also observed. The 20 viral miRNAs detected from PK15 cell line by using Illumina deep sequencing were derived from the open reading frame (ORF), IRS, and TRS regions of the PRV genome [81].

Host miRNAs, like viral miRNAs, can participate in the regulation of virus infection. Many studies have shown that PRV infection can affect the expression of miRNAs in host cells [76,77,81,82]. Members of the miR-146 family can regulate host inflammatory and immune responses [83,84]. In PRV infected mouse neuroblastoma cells, miR-146b-5p was significantly up-regulated after PRV infection, which could promote PRV replication and negatively regulate type I interferon response [82]. Furthermore, other host miRNAs can also target multiple viral genes such as miR-1249-3p, miR-6538, miR-466k, and miR-714 to regulate the PRV infection [82].

### 4.2. Other Non-Coding RNAs Transcribed by LAT Gene

Long non-coding RNA (lncRNA) has the function of regulating gene expression [67]. In latently infected ganglia, the LAT gene of HSV can express different sizes of lncRNAs. They can accumulate in the nucleus of latently infected neurons to induce the formation of facultative heterochromatin, promote lytic gene silencing, and evade the host immune response [66,70]. PRV can up-regulate the expression of host lncRNAs in infected cells, thus promoting the replication of itself [85].

Peng et al. identified two ncRNAs in the first 1.5 kb LAT region of HSV which were different from the typical miRNA structure [86]. They possessed the feature of sRNA, so the two ncRNAs were called LAT sRNA1 and sRNA2.The first 1.5 kb LAT region of HSV plays an important role in suppressing productive infection, resisting apoptosis, maintaining latent infection, and ensuring a high rate of viral reactivation [87,88]. According to the complementary pairing of the two sRNAs with ICP4 mRNA and the position of their corresponding DNA sequence, it is speculated that sRNA1 and sRNA2 could inhibit the translation of ICP4 mRNA and apoptosis. Shen et al. confirmed this speculation [69]. ICP4 is essential for productive infection and viral reactivation. The LAT sRNA2 can inhibit the translation of ICP4 mRNA, so the functions of the latter can be affected in regulating the virus infection. LAT sRNA1 has a stronger ability to suppress productive infection than sRNA2, but it has no significant effect on the expression of ICP0 or ICP4 protein. It is predicted that LAT sRNA1 may target mRNA of VP16 and UL8 to inhibit the production of regulatory proteins necessary for viral replication. Single point mutations in LAT sRNA1 and sRNA2 can reduce the ability of LAT to inhibit apoptosis [69]. Therefore, sRNA1 and sRNA2 take an important role in the anti-apoptotic effect of LAT.

In some studies, LAT sRNA1 and sRNA2 are named LAT sncRNA1 and LAT sncRNA2, respectively. LAT sncRNA1 and sncRNA2 can regulate the signal pathway of retinoic acid-induced gene I (RIG-I) to improve cell survival [68]. Herpesvirus entry medium (HVEM) is a cell surface protein that mediates the attachment and entry of HSV into cells. HVEM can regulate the cellular immune response and inhibit apoptosis. In latently infected neurons, LAT can up-regulate the expression of HVEM through the interaction of LAT sncRNA1 and sncRNA2 with the HVEM promoters, thereby promoting cell survival and helping the virus escape host immunity [89].

### 4.3. LAT Encoding Protein in Latent Infection

Open reading frames (ORFs) are DNA sequences that are capable of encoding proteins. Eight potential ORFs exist in the first 1.5 kb LAT of HSV-1. Introducing point mutations into the ATG of ORFs can reduce the activity of LAT in inhibiting apoptosis [90]. Although no bioactive LAT encoding protein was detected in latently infected neurons, its biological function was confirmed in vitro [91]. HSV-1 LAT encoding protein can restore the replication level of the ICP0 gene-deficient strain in vitro and improve the growth level of the virus in vivo. Therefore, it was implied that the LAT encoding protein might have functions similar to ICP0 in the reactivation of virus latent infection [91,92].

## 5. Effect of Pseudorabies Vaccine on Virus Latent Infection

The PRV vaccine could not prevent the establishment of latent infection of wild-type strains. As early as 1981, it was reported that latent infection of PRV still existed in pig farms vaccinated with live attenuated PRV vaccine [93]. Since then, a large number of studies have confirmed that the clinical symptoms and mortality of infected pigs can be reduced by active immunization of the inactivated vaccine, live attenuated vaccine, and subunit vaccine or passive immunization of maternal antibody, but latent infection of the virulent PRV strain cannot be prevented in the PNS of the host [18,26,94,95,96,97,98]. Inoculation with the PRV vaccine cannot prevent the virulent strain from establishing latent infection in host pigs, but it can reduce the amount and time of virus excretion after the virulent strain reactivated [99].

Live attenuated PRV vaccines can also establish latent infections in PNS of pigs like virulent strains [100]. The latent infection of the PRV gene deletion vaccine can affect that of wild-type strains, and there is a significant negative correlation between them [101]. By increasing the level of latent infection of the gene deletion vaccine, we can reduce or eliminate that of wild-type PRV in the host. However, one problem that needs to be considered is genetic recombination. Virus recombination is affected by the dose of the inoculated virus, the time interval between the two viruses, the distance between marker mutations, genetic homology, virulence, and latency [102,103,104]. Homologous recombination often occurs among the same *Alphaherpesvirinae* [104]. The PRV gene deletion vaccine may recombine with the wild-type virus strain to produce a highly virulent variant strain, or a variant strain that cannot be differentiated by serological method [105]. To avoid this, the biosafety of live attenuated vaccines must be carefully evaluated. The LAT gene is a nonessential factor for the establishment of viral latent infection, but it plays a pivotal role in the reactivation of the virus [58]. Mahjoub et al. obtained a PRV mutant with 9 LAT miRNAs deletions. This mutant could establish latent infection, but it could not be reactivated [106]. In HSV, the deletion of LAT fragments can affect the reactivation rate, the virulence, and the ability of anti-apoptosis [74,107,108]. The specific molecular mechanism of LAT still needs to be explored. LAT plays a key role in regulating the conversion between virus latency and reactivation, inhibiting apoptosis to prolong the survival time of neurons. Perhaps we could take some measures to modify the LAT gene to develop a non-reactivated PRV vaccine.

## 6. Conclusions and Perspectives

In this review, we summarize the characteristics of PRV latent infection, the transcriptional characteristics and functions of LAT gene, and the effects of PRV vaccine on the establishment of latent infection of virulent PRV strains. Studying LAT ncRNA can provide a new perspective for elucidating the molecular mechanism of PRV latent infection. We can also take some effective measures to control the latent infection of wild-type PRV in pigs, such as developing effective vaccines or drugs to inhibit the establishment or reactivation of latent infection of wild type PRV. During latent infection of PRV, no infectious virions are produced, but the genome of the virus can be detected in the host. However, latent infection of the virus can be detected by serology, in situ hybridization, tissue co-culture, polymerase chain reaction (PCR), fluorescence quantitative PCR, and real-time recombinant enzyme-assisted amplification [46,109,110,111,112,113]. Similar to the PRV virulent strain, the PRV gene deletion vaccine can also establish latent infection in the PNS of pigs. When pigs are inoculated with two vaccines of different gene deletions, genetic recombination may occur between them, and then new strains may be produced. Therefore, only one gene deletion vaccine is recommended in the same pig farm or the same animal individual to avoid genetic recombination between vaccine strains [102,103].

The LAT gene is closely related to latent infection. The latent infection of the PRV gene deletion vaccine was negatively correlated with that of wild-type PRV strains, and the LAT gene could affect the reactivation rate of the virus. Therefore, we can modify the LAT gene to develop a genetic engineering vaccine to restrict the latent infection of PRV wild-type strain in pigs. The ncRNAs transcribed by the LAT gene play an important role in the process of viral latent infection. Therefore, RNA interference and RNA silencing are applied to these ncRNAs to regulate the latency and reactivation of PRV. At present, there is little research on the ncRNA of PRV LAT. Therefore, in order to further understand the molecular mechanism of PRV latent infection, we can emphasize the study of ncRNAs produced by PRV LAT.

PRV can be transmitted from pigs to humans, threatening human health [28,29,30,31,32,33,34,35,36,37,38,39]. PRV infection can cause fever, headache, endophthalmitis, and acute encephalitis [28,29,30,31,32,33,34,35,36,37,38,39]. Antiherpes drugs such as valaciclovir, penciclovir, and phosphonoformate can control the symptoms of patients, but the visual and nerve damage caused by PRV is irreversible. PRV can establish reactivatable latent infection in mice [16,114,115]. Humans are non-natural hosts like mice, so it is possible for PRV to establish reactivatable latent infection in the body, resulting in irregular recurrence of the disease. Therefore, the subsequent progress of PRV-infected patients should be tracked in the long-term to evaluate the probability. If the molecular mechanism of latent infection is clear, some specific targets may be found, and then relevant drugs can be developed to block the latency and reactivation of PRV.

## Figures and Tables

**Figure 1 viruses-14-01379-f001:**
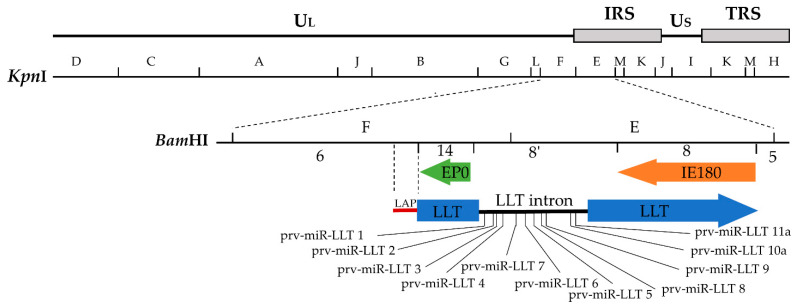
Schematic of the PRV genome and location of the LAT gene. The images from top to bottom show the structure of the PRV genome, the *Kpn*I restriction enzyme map, the *Bam*HI restriction enzyme map, the transcription location and direction of LAT, IE180, and EP0 genes, the LAP location, and the location of prv-miR-LLT 1-11 in the LLT intron.

## Data Availability

Not applicable.

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
