# Peer review of "The Role of Latency-Associated Transcripts in the Latent Infection of Pseudorabies Virus"

_viruses, 2022, doi:10.3390/v14071379_

Round 1

Reviewer 1 Report

In this article, “Deng et alreviewed the characteristics of pseudorabies virus (PRV) latent infections and the transcriptional characteristics of the LAT genes. Then, they analyzed the functions of non-coding RNA (ncRNA) produced by LAT genes and their importance in latent infections. Also, they discussed about the zoonotic ability of PRV. This point (even if a little bit out of scope) needs to be further discussed to help the reader to really figure out about the risk for pig breeders, veterinarians (number of cases…) … Overall this review article is interesting and quite well-written.

Strengths: Interesting subject and quite well-written

Weaknesses: English could be improved. Indeed, the meaning of some sentences is difficult to get. Figures and or tables are missing, please add some. Recombination should be further discussed, especially if a new generation of live vaccine could reach the market.  

Please find below a detailed list of my moderate remarks/suggestions.

Major

-No table, no figure (about latency for instance) … This is missing.

Moderate

-L33: Please replace the “crazy” by something more suitable. In general, English could be improved (see also L80, L83 “eliminated the inhibition”, L141, L211, L256). The meaning of some sentences is unclear.

-L40: The features of these mutant strains could be more detailed.

-L45-50: Can we really conclude at this stage about the zoonotic potential? That section could be more detailed (number of cases…).

-L62-63: Please add a reference.

-L78: In which context? Latent infection I suppose. What is the situation in the case of productive infection regarding that type of inhibition?

-L93: Please provide more details and add a reference.

-L143-144: Please add a reference.

-L153: Please specify which cell protein.

-L168: Even the detection by Pattern Recognition Receptors (PRRs)? Please specify.

-L170: “more advantages” such as? Please specify.

-L177: However, genome organization is quite different between PRV and HSV-1. Some Alphaherpesvirinae are closer to PRV in terms of genome organization than HSV-1. See for instance BoHV-1, VZV… The switch from PRV to HSV-1 is not always easy to follow and some clarifications could be useful regarding that specific point.

-L189-190: Please add a reference.

-L207-210: Please add references.

-L221: “thought”, based on what? Please add.

-L223: PK15 cells, a quite artificial system, what about other cell lines or primary cells.

-L270: Is there an explanation for this absence of detected proteins? Please specify.

-L291: Some studies and review articles about recombination in bovine herpesvirus 1 and alphaherpesviruses in general would be helpful here. Recombination could more detailed and discussed. Please see for reviews: doi: 10.1016/j.vetmic.2005.11.012, doi: 10.1002/rmv.451 and original article: doi: 10.1128/jvi.77.23.12535-12542.2003, doi: 10.1128/jvi.78.8.3872-3879.2004, doi: 10.1128/JVI.78.18.9828-9836.2004.

-L310-311: This point could be more detailed.

-L319-321: Not recommended by who? Could you please develop.

-L325-327: The sentence is not totally clear. Moreover, it could bring new risks.

-L333-341: This section is quite repetitive and a more general discussion is missing.

-L333: References should be added again here.

Minor

-L13: Please specify “the only active region of what”.

-L18: “the” can be removed before possible.

-L24: “Herpesviridae” must be in italic (Latin), as well as all the “in vivo”, “in vitro”, “in situ” …

-L25: Please replace alpha-herpesvirus by Alphaherpesvirinae in italic.

-L35: Please add “consequently” before “latent infection”.

-L53: PNS, please specify where already here.

-L87: I would replace “detect” by “house”, “contain” or something else.

-L111: Please provide the exact definition of EP0 acronym and add a space before the bracket.

-L138: Please replace “a necessary molecule” by is not “required” for…

-L157: Please replace “are” by “were”.

-L182: Please replace “destroying” by something like “affecting” or “altering”.

-L248: Please remove “quickly”.

-L253: “at mRNA”, not clear…

-L260: In which context? During infection I suppose.

-L283: The “fortunately” is weird here, not clear to what it is referring.

-L303: “can” could be replaced by “could”.

-L307: Please add a “s” to function.

-L313: “Therefore” is quite problematic here and should be replaced by “However”.

-L342: “can be found” or not…

Reviewer 2 Report

This is a write-written review paper on the role of latency-associated transcripts and non-coding RNA in latency process during pseudorabies virus (PRV) infection in pigs. Meanwhile, pseudorabies has been efficiently controlled in a number of EU countries by DIVA vaccination strategy, in Asia the disease seems to reemerge. The virulence of observed viruses increased apparently also. The authors aimed to discuss the potential role of LAT transcripts and ncRNAs in latency stage of PRV infection. The paper presents an interesting review of current state of knowledge focusing alpha-herpesviruses of swine and the latency.

Round 2

Reviewer 1 Report

The article has been well-improved.

Please make sure all the abbreviations are defined, especially in the figure legend where there are currently not.

Author Response

Dear reviewer,

Thanks for your comments concerning our manuscript entitled “The role of Latency-associated transcripts in the latent infection of pseudorabies virus” (Manuscript ID: viruses-1772598). 

Please see below, in red, for a point-by-point response to the reviewers’ comments.

Point 1: Please make sure all the abbreviations are defined, especially in the figure legend where there are currently not.

Response 1: Thank you for this comment. We have revised the manuscript conscientiously and all abbreviations have been defined.

We appreciate your warm work earnestly and hope that the corrections will meet with approval.

Once again, thank you very much for your comments and suggestions.

Thank you and best regards.

Yours Sincerely,

Jiahuan Deng
E-mail: 13420690147@163.com

Corresponding author:
Name: Chunmei Ju
E-mail: juchunmei@scau.edu.cn